# Siegel Neural Networks

**Xuan Son Nguyen    Aymeric Histace    Nistor Grozavu**
ETIS, UMR 8051, CY Cergy Paris University, ENSEA, CNRS
`{xuan-son.nguyen,aymeric.histace}@ensea.fr`
`nistor.grozavu@cyu.fr`

## Abstract

Riemannian symmetric spaces (RSS) such as hyperbolic spaces and symmetric positive definite (SPD) manifolds have become popular spaces for representation learning. In this paper, we propose a novel approach for building discriminative neural networks on Siegel spaces, a family of RSS that is largely unexplored in machine learning tasks. For classification applications, one focus of recent works is the construction of multiclass logistic regression (MLR) and fully-connected (FC) layers for hyperbolic and SPD neural networks. Here we show how to build such layers for Siegel neural networks. Our approach relies on the quotient structure of those spaces and the notation of vector-valued distance on RSS. We demonstrate the relevance of our approach on two applications, i.e., radar clutter classification and node classification. Our results successfully demonstrate state-of-the-art performance across all datasets.

## 1   Introduction

Deep neural networks are generally built upon the assumption that the data or features at hand exhibit Euclidean latent structure. Unfortunately, this assumption does not hold in many applications [1, 16] where the data or features lie on a multidimensional curved surface which is locally Euclidean. For such applications, those models often produce unsatisfactory results because their building blocks based on Euclidean geometry break the geometric stability principle that plays a crucial role in geometric deep learning architectures [8]. To deal with this issue, many Riemannian neural networks have been developed for solving a wide variety of machine learning problems [16, 22, 25, 29, 30, 32]. In this paper, we restrict our attention to discriminative neural networks with manifold-valued output.

Early works focus either on hyperbolic spaces [16, 38] or on matrix manifolds [20, 21, 22]. In an attempt to develop a unified framework for a more general setting, the authors of [31, 33, 34] leverage the gyro-structure of certain Riemannian manifolds. However, in the general case, their methods cannot provide an explicit form for the point-to-hyperplane distance which is at the heart of their proposed network building blocks [38] since the distance must be derived with respect to a specific Riemannian metric. The work in [35] alleviates this issue by deriving a closed form for the point-to-hyperplane distance associated with $G$-invariant Riemannian metrics on RSS. Although this work is applicable to Siegel spaces, the construction of the MLR and FC layers [35] from the derived distance is heavily based on the maximal abelian subspaces of those spaces. This can affect the ability of the resulting networks to learn rich representations and complex decision boundaries.

In this paper, we propose a novel approach for building neural networks on Siegel spaces. Those are among the most versatile RSS [27] and have many attractive theoretical properties. The two well-established models of Siegel spaces, i.e., the Siegel upper space and the Siegel disk [39] generalize the complex Poincaré upper plane and the complex Poincaré disk [18] to spaces of symmetric complex matrices [36]. SPD manifolds associated with affine-invariant Riemannian metrics [37] are also special cases of Siegel spaces associated with Siegel metrics [39]. Despite the potential of Siegel spaces in capturing rich geometrical structures, they are much less studied in the context

of deep learning compared to other RSS. Although few recent works [27, 41] use Siegel spaces as representation spaces, they only focus on learning and visualizing embeddings in natural language processing and graph tasks. Therefore, an effective framework for building discriminative Siegel neural networks is still missing. In summary, our contributions are the following:

- We propose a novel formulation of MLR layers for Siegel neural networks based on a gyrovector approach which has proven effective in building hyperbolic and SPD neural networks [16, 31, 38]. This formulation is then extended to the product space setting.

- We show that the notion of vector-valued distance [24] which captures the complete $G$-congruence invariant of pairs of points on RSS enables another formulation of MLR layers for Siegel neural networks in a natural way. This formulation leads to more compact MLR layers than those obtained by the first formulation.

- We introduce two variants of FC layers for Siegel neural networks.

- We build the first discriminative Siegel neural networks and evaluate them on the radar clutter classification and node classification tasks.

## 2    Mathematical Background

### 2.1    Siegel Spaces

The Siegel upper space $\mathbb{SH}_m$ is defined as
$$\mathbb{SH}_m = \{x = u + iv : u \in \mathrm{Sym}_m, v \in \mathrm{Sym}_m^+\},$$
where $\mathrm{Sym}_m$ and $\mathrm{Sym}_m^+$ denote the space of $m \times m$ real symmetric matrices and that of $m \times m$ SPD matrices, respectively.

Another model for Siegel spaces is the Siegel disk defined by
$$\mathbb{SD}_m = \left\{w \in \mathrm{Sym}_{m,\mathbb{C}} : I_m - ww^H \in \mathbb{H}_m^+\right\},$$
where $I_m$ is the $m \times m$ identity matrix, $\mathrm{Sym}_{m,\mathbb{C}}$ and $\mathbb{H}_m^+$ denote the space of $m \times m$ complex symmetric matrices and that of $m \times m$ Hermitian positive definite (HPD) matrices, respectively, and $w^H$ is the conjugate transpose of $w$.

One can convert a point $x \in \mathbb{SH}_m$ to $\mathbb{SD}_m$ using the matrix Cayley transformation defined as
$$\varphi(x) = (x - iI_m)(x + iI_m)^{-1}.$$

The inverse matrix Cayley transformation that converts a point $w \in \mathbb{SD}_m$ to $\mathbb{SH}_m$ is given by
$$\varphi^{(-1)}(w) = i(I_m + w)(I_m - w)^{-1}.$$

In the following, we shall focus on the Siegel upper space model.

**Quotient Structure**    Denote by
$$\mathrm{Sp}_{2m} = \left\{\begin{bmatrix} a & b \\ c & d \end{bmatrix} : ab^T = ba^T, cd^T = dc^T, ad^T - bc^T = I_m\right\}$$
the real symplectic group. This group acts transitively on $\mathbb{SH}_m$ by the action $s[x] = (ax + b)(cx + d)^{-1}$, where $s = \begin{bmatrix} a & b \\ c & d \end{bmatrix} \in \mathrm{Sp}_{2m}$ and $x \in \mathbb{SH}_m$. The stabilizer group of $x = iI_m \in \mathbb{SH}_m$ is the subgroup of symplectic orthogonal matrices $\mathrm{SpO}_{2m}$ defined as:
$$\mathrm{SpO}_{2m} = \left\{\begin{bmatrix} a & b \\ -b & a \end{bmatrix} : a^T a + b^T b = I_m, a^T b \in \mathrm{Sym}_m\right\} = \mathrm{Sp}_{2m} \cap \mathrm{O}_{2m},$$
where $\mathrm{O}_{2m}$ is the group of orthogonal matrices. We thus have the identification $\mathbb{SH}_m \cong \mathrm{Sp}_{2m} / \mathrm{SpO}_{2m}$. The element in $\mathrm{Sp}_{2m}$ that transforms $iI_m$ to $x = u + iv \in \mathbb{SH}_m$ via the group action is given by the map $\phi(\cdot)$ in the following identification:
$$\psi : \mathbb{SH}_m \to \mathrm{Sp}_{2m} / \mathrm{SpO}_{2m}$$
$$x \mapsto \begin{bmatrix} v^{\frac{1}{2}} & uv^{-\frac{1}{2}} \\ \mathbf{0} & v^{-\frac{1}{2}} \end{bmatrix} \mathrm{SpO}_{2m} = \phi(x)\,\mathrm{SpO}_{2m}.$$

**Riemannian Metric** The Riemannian metric (also referred to as Symplectic metric) for the Siegel upper space model is given [39] by

$$ds_x^2 = 2\operatorname{Tr}(v^{-1}dxv^{-1}d\bar{x}), \tag{1}$$

where $x = u + iv \in \mathbb{SH}_m$, and $\operatorname{Tr}(\cdot)$ is the matrix trace (see Appendix 3.2 for further discussions). The associated Riemannian distance $d_{\mathbb{SH}}(x, y)$ between two points $x, y \in \mathbb{SH}_m$ is given by

$$d_{\mathbb{SH}}(x, y) = \sqrt{\sum_{j=1}^m \log^2 \left( \frac{1 + r_j^{\frac{1}{2}}}{1 - r_j^{\frac{1}{2}}} \right)},$$

where $r_j, j = 1, \ldots, m$ are the eigenvalues of the cross-ratio $R(x, y)$ defined as

$$R(x, y) = (x - y)(x - \bar{y})^{-1}(\bar{x} - \bar{y})(\bar{x} - y)^{-1},$$

where $\bar{x}$ denotes the complex conjugate of $x$.

## 2.2 Riemannian Symmetric Spaces

This section briefly reviews key concepts from the theory of RSS for our work. We refer the interested reader to Appendix 3.1 for further discussions.

A symmetric space is a connected Riemannian manifold $X$ with a geodesic-reversing isometry at each point. In other words, for each point $x \in X$ there is an isometry $\sigma_x$ of $X$ such that $\sigma_x(x) = x$ and the differential of $\sigma_x$ at $x$ is multiplication by $-1$ [7]. Siegel spaces belong to a family of RSS referred to as symmetric spaces of noncompact type or noncompact RSS. In the following, we refer to noncompact RSS as RSS or symmetric spaces. Let $G$ be a connected noncompact semisimple Lie group with finite center, and let $K$ be a maximal compact subgroup of $G$. Then a symmetric space $X$ consists of the left cosets

$$X := G/K := \{x = gK | g \in G\}.$$

The action of $G$ on $X = G/K$ is defined as $g[x] = g[hK] = ghK$ for $x = hK \in X$, $g, h \in G$. Let $o$ be the origin $K$ in $X$, then the map $\gamma : gK \mapsto g[o]$ is a diffeomorphism of $G/K$ onto $X$.

Let $d(.,.)$ be the distance induced by the Riemannian metric. A *geodesic ray* in $X$ is a map $\delta : [0, \infty) \to X$ such that $d(\delta(t), \delta(t')) = |t - t'|, \forall t, t' \geq 0$. A *geodesic line* in $X$ is a map $\delta : \mathbb{R} \to X$ such that $d(\delta(t), \delta(t')) = |t - t'|, \forall t, t' \in \mathbb{R}$.

The geometry of $X$ can be studied through the geometry of its *maximal flats* [2, 7, 18, 24]. A subspace $F \subset X$ is called a flat of dimension $k$ (or a $k$-flat) if it is isometric to $\mathbb{R}^k$. The subspace $F$ is called a maximal flat if it is not contained in any flat of bigger dimension. Since all maximal flats in $X$ are isometric [18], they can be simultaneously identified with a model (maximal) flat $F_{mod}$.

Flats are decomposed into *Weyl chambers*. A Weyl chamber in a maximal flat $F$ with tip at $x \in F$ is a connected component of the set of points $x' \in F \setminus \{x\}$ such that the geodesic line through $x$ and $x'$ is contained in a unique maximal flat [7]. Since $G$ acts transitively on the set of Weyl chambers in $X$ [18], they can be simultaneously identified with a Weyl chamber $\Delta$. The subgroup of isometries of $F$ which are induced by elements of $G$ is isomorphic to a semidirect product $\mathbb{R}^r \rtimes W$. $W$ is called the *Weyl group* of $G$ and $X$.

Any symmetric space $X$ is associated with a *boundary at infinity* $\partial X$ constructed as the set of equivalence classes of geodesic rays in $X$. Two rays are considered equivalent if their images are a bounded distance apart [7]. The equivalence class of a geodesic ray $\delta$ is denoted by $\delta(\infty)$.

## 3 Proposed Approach

Our proposed point-to-hyperplane distances based on the quotient structure of Siegel spaces and the vector-valued distance are presented in Sections 3.1 and 3.2, respectively. In Section 3.3, we present our MLR models and introduce two variants of FC layers for Siegel neural networks. In our work, we focus on Siegel spaces but many of our results can also be stated for other RSS. To simplify the notation, we use the letters $X$, $G$, and $K$ (see Section 2.2) to denote the spaces associated with Siegel spaces (see Section 2.1) unless otherwise stated.

### 3.1 Point-to-hyperplane Distances Based on the Quotient Structure of Siegel Spaces

#### 3.1.1 Hyperplanes

We start with a formulation of Euclidean hyperplanes given as

$$\mathcal{H}_{a,b}^E = \{x \in \mathbb{R}^m : \langle x, a \rangle - b = 0\},$$

where $a \in \mathbb{R}^m \setminus \{\mathbf{0}\}$, $b \in \mathbb{R}$, and $\langle \cdot, \cdot \rangle$ is the Euclidean inner product. Hyperplane $\mathcal{H}_{a,b}^E$ can be reformulated [16] as

$$\mathcal{H}_{a,b}^E = \{x \in \mathbb{R}^m : \langle -p + x, a \rangle = 0\}, \tag{2}$$

where $p \in \mathbb{R}^m$ and $\langle p, a \rangle = b$.

To generalize Euclidean hyperplanes to our setting, we follow the approach in [33, 34] which relies on a binary operation, an inverse operation, and an inner product defined on the target space. Let $x = gK, y = hK \in X$ where $g, h \in G$. In the case of $\mathbb{SH}_m$, $g = \phi(x), h = \phi(y)$ where $\phi(u + iv) = \begin{bmatrix} v^{\frac{1}{2}} & uv^{-\frac{1}{2}} \\ \mathbf{0} & v^{-\frac{1}{2}} \end{bmatrix}, u + iv \in \mathbb{SH}_m$.

**Definition 3.1** ([35])**.** *The binary operation $\oplus$ and inverse operation $\ominus$ are defined as*

$$x \oplus y = ghK, \quad \ominus x = g^{-1}K.$$

We propose the following inner product.

**Definition 3.2.** *The inner product $\langle \cdot, \cdot \rangle_{\mathbb{S}}$ on $X$ is defined as*

$$\langle x, y \rangle_{\mathbb{S}} = \langle \log(gg^T), \log(hh^T) \rangle,$$

*where $\log(\cdot)$ denotes the matrix logarithm.*

Our proposed inner product is motivated by Proposition 3.3. (see Appendix 4.1 for its proof).

**Proposition 3.3.** *The inner product $\langle \cdot, \cdot \rangle_{\mathbb{S}}$ agrees with the Riemannian distance, i.e.,*

$$\| \ominus x \oplus y \|_{\mathbb{S}} \propto d_{\mathbb{SH}}(x, y),$$

*where $x, y \in X$, and the norm $\| \cdot \|_{\mathbb{S}}$ is induced by the inner product $\langle \cdot, \cdot \rangle_{\mathbb{S}}$. Furthermore, the inner product $\langle \cdot, \cdot \rangle_{\mathbb{S}}$ is invariant under the action of K, i.e., for any $k \in K$,*

$$\langle x, y \rangle_{\mathbb{S}} = \langle k[x], k[y] \rangle_{\mathbb{S}}.$$

Note that both properties in Proposition 3.3 are satisfied by the inner products in [33, 35] and the second property is also satisfied by the one in [19]. Note also that these properties hold for the more general case in which $G$ is the general linear group or its subgroup, and $K$ is the group of orthogonal matrices or its subgroup (see Appendix 4.1). We are now ready to define hyperplanes.

**Definition 3.4.** *Let $a, p \in X$. Then hyperplanes on $X$ are defined as*

$$\mathcal{H}_{a,p} = \{x \in X : \langle \ominus p \oplus x, a \rangle_{\mathbb{S}} = 0\}.$$

Segments of the form $\ominus p \oplus x$ can be regarded as Siegel analogs of Euclidean lines. Thus, $\mathcal{H}_{a,p}$ has a similar interpretation as a Euclidean hyperplane, i.e., the former contains a fixed point $p \in X$ and any point $x \in X$ such that the segment $\ominus p \oplus x$ is orthogonal to a fixed direction $a$. Therefore, hyperplanes as given in Definition 3.4 are natural extensions of Euclidean hyperplanes.

#### 3.1.2 Point-to-hyperplane Distance

The distance $\bar{d}(x, \mathcal{H}_{a,p})$ between a point $x \in X$ and a hyperplane $\mathcal{H}_{a,p}$ given in Definition 3.4 can be formulated [33] as

$$\bar{d}(x, \mathcal{H}_{a,p}) = \sin(\angle xp\bar{q})d(x, p),$$

where $\angle xp\bar{q}$ is the gyroangle [33, 43] (see Appendix 3.4) between $\ominus p \oplus x$ and $\ominus p \oplus \bar{q}$, and $\bar{q}$ is computed as

$$\bar{q} = \arg\max_{q \in \mathcal{H}_{a,p} \setminus \{p\}} \left( \frac{\langle \ominus p \oplus q, \ominus p \oplus x \rangle_{\mathbb{S}}}{\| \ominus p \oplus q \|_{\mathbb{S}} \| \ominus p \oplus x \|_{\mathbb{S}}} \right),$$

By convention, $\sin(\angle xpq) = 0$ for any $x, q \in \mathcal{H}_{a,p}$. Theorem 3.5 gives a closed form for the point-to-hyperplane distance on Siegel spaces (see Appendix 4.2 for its proof).

**Theorem 3.5.** *Let $x, a, p \in X$ and let $\mathcal{H}_{a,p}$ be a hyperplane as given in Definition 3.4. Then*

$$\bar{d}(x, \mathcal{H}_{a,p}) = \frac{|\langle \log(\phi(p)^{-1}\phi(x)\phi(x)^T\phi(p)^{-T}), \log(\phi(a)\phi(a)^T)\rangle|}{\|\log(\phi(a)\phi(a)^T)\|},$$

*where $\|\cdot\|$ denotes the Euclidean norm, and the map $\phi(\cdot)$ is given in Section 2.1.*

### 3.1.3 Product Spaces

We now extend the above method to the product space setting. Let $X$ be defined as the Cartesian product $X = X_1 \times \ldots \times X_L$, where $X_j = G_j/K_j, j = 1, \ldots, L$ are RSS, $G_j$ is a connected noncompact semisimple Lie group with finite center, $K_j$ is a maximal compact subgroup of $G_j$. Here we focus on the Cartesian product of SPD and Siegel spaces. Each point $x \in X$ can be described through its coordinates $x = (x_1, \ldots, x_L), x_j \in X_j, j = 1, \ldots, L$. In this setting, one has simple decompositions of the tangent space, the exponential map, and the squared Riemannian distance [15, 17, 42]. When $X_j$ is an SPD space, $G_j$ is the general linear group and $K_j$ is the group of orthogonal matrices (see Appendix 3.3). Thus one can define the binary operation, inverse operation, and inner product on $X_j$ as in Definitions 3.1 and 3.2, and the results in Proposition 3.3 still hold. By abuse of notation, we shall use the same notations for those operations as in Section 3.1.1.

**Definition 3.6.** *Let $x = (x_1, \ldots, x_L)$, $y = (y_1, \ldots, y_L) \in X, x_j, y_j \in X_j, j = 1, \ldots, L$. The binary operation $\oplus$ and inverse operation $\ominus$ on $X$ are defined as*

$$x \oplus y = (x_1 \oplus y_1, \ldots, x_L \oplus y_L), \ \ominus x = (\ominus x_1, \ldots, \ominus x_L).$$

**Definition 3.7.** *The inner product $\langle \cdot, \cdot \rangle_\mathbb{S}$ on $X$ is defined as*

$$\langle x, y \rangle_\mathbb{S} = \sum_{j=1}^{L} \langle x_j, y_j \rangle_\mathbb{S}.$$

The following theorem (see Appendix 4.3 for its proof) extends Theorem 3.5 to the considered setting.

**Theorem 3.8.** *Let $\mathcal{H}_{a,p}$ be a hyperplane as given in Definition 3.4, where $a = (a_1, \ldots, a_L), p = (p_1, \ldots, p_L), a_j = w_j K_j, p_j = h_j K_j \in X_j, w_j, h_j \in G_j, j = 1, \ldots, L$, and let $x = (x_1, \ldots, x_L) \in X$ where $x_j = g_j K_j \in X_j, g_j \in G_j$. Then*

$$\bar{d}(x, \mathcal{H}_{a,p}) = \frac{|\sum_{j=1}^{L} \langle \log(h_j^{-1} g_j g_j^T h_j^{-T}), \log(w_j w_j^T)\rangle|}{\sqrt{\sum_{j=1}^{L} \|\log(w_j w_j^T)\|^2}}.$$

## 3.2 Point-to-hyperplane Distances Based on Vector-Valued Distances

As shown in [38], the formulation of Euclidean hyperplanes in Eq. (2) has an over-parameterization issue, i.e., it increases the number of parameters from $m+1$ to $2m$ in each class. Our formulation of hyperplanes in Section 3.1.1 (see Definition 3.4) follows that formulation and thus suffers from a similar issue. In this section, we propose another method for constructing the point-to-hyperplane distance which results in more compact MLR layers for Siegel neural networks.

### 3.2.1 Hyperplanes

We start with a similar formulation of Euclidean hyperplanes in Eq. (2) but use a different parameterization. Given $p \in \mathbb{R}^m$ and $\xi \in \partial \mathbb{R}^m$, the Euclidean hyperplane $\mathcal{H}_{\xi,p}^E$ parameterized by $p$ and $\xi$ can be defined [35] by

$$\mathcal{H}_{\xi,p}^E = \{x \in \mathbb{R}^m : \langle p - x, a \rangle = 0\} = \{x \in \mathbb{R}^m : \langle \mathrm{vec}(x, p), a \rangle = 0\},$$

where $\xi$ is the equivalence class of the geodesic ray $\delta(t) = t\frac{a}{\|a\|}, a \in \mathbb{R}^m \setminus \{\mathbf{0}\}$, and the function $\mathrm{vec}(x, p) = p - x$ denotes the translation carrying $x$ to $p$.

In a symmetric space, a natural analog of the function $\mathrm{vec}(\cdot, \cdot)$ is the vector-valued distance function [23, 24]. Given two points $x, y \in X$, one computes a $G$-invariant distance by first transforming (via the $G$-action) $x$ and $y$ to $x'$ and $y'$ on the model flat $F_{mod}$, respectively, and then identifying the

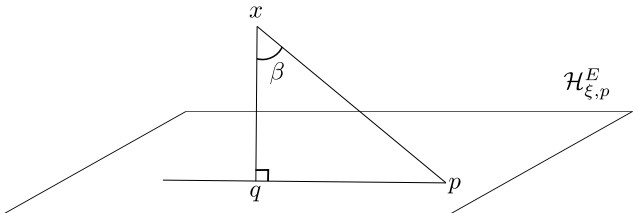

Figure 1: The distance between a point $x \in \mathbb{R}^m$ and a Euclidean hyperplane $\mathcal{H}^E_{\xi,p}$.

translation modulo the action of the Weyl group carrying $x'$ to $y'$. Note that in $\mathbb{R}^m$, the projections of $x$ and $p$ on a maximal flat are precisely $x$ and $p$, respectively. The domain of the resulting distance function, which is a fundamental domain for the action of the Weyl group on the translations, can be canonically identified with the Weyl chamber $\Delta$. The above observation motivates the following definition.

**Definition 3.9.** *Let $d_\Delta(\cdot,\cdot) : X \times X \to \Delta$ be the vector-valued distance function on $X$. Let $p \in X$, $\xi \in \partial X$, and let $a_\xi \in \Delta$ be such that $\xi$ is the equivalent class of the geodesic ray $\delta(t) = k\exp(ta_\xi)K, k \in K$. Then hyperplane $\mathcal{H}_{\xi,p}$ is defined as*

$$\mathcal{H}_{\xi,p} = \{x \in X : \langle d_\Delta(x,p), a_\xi \rangle = 0\}.$$

A hyperplane given in Definition 3.9 has a clear interpretation, i.e., it contains a fixed point $p \in X$ and any point $x \in X$ such that the vector-valued distance between $x$ and $p$ is orthogonal to a fixed direction $a_\xi$. We note that the notion of vector-valued distance has been employed in [27, 28] for learning and visualizing embeddings in natural language processing and graph tasks. However, none of those works reveals the analogies discussed above for defining Siegel hyperplanes.

### 3.2.2 Point-to-hyperplane Distance

Let $p \in \mathbb{R}^m$ and $\xi \in \partial \mathbb{R}^m$. The distance $\bar{d}(x, \mathcal{H}^E_{\xi,p})$ between a point $x \in \mathbb{R}^m$ and hyperplane $\mathcal{H}^E_{\xi,p}$ can be computed (see Fig. 1) as

$$\bar{d}(x, \mathcal{H}^E_{\xi,p}) = d(x,p)\cos(\beta),$$

where $\beta$ is the angle between the segments $[x,p]$ and $[x,q]$, and $q$ is the projection of $x$ on $\mathcal{H}^E_{\xi,p}$. By convention, $\bar{d}(x, \mathcal{H}^E_{\xi,p}) = 0, \forall x \in \mathcal{H}^E_{\xi,p}$. The above equation can be rewritten as

$$\bar{d}(x, \mathcal{H}^E_{\xi,p}) = d(x,p)\cos\angle_x(p,\xi),$$

where $\angle_x(p,\xi)$ denotes the angle at $x$ between the geodesic segment $[x,p]$ and the geodesic ray which issues from $x$ and is in the class $\xi$. We generalize the above equation to our setting.

**Definition 3.10.** *Let $p \in X$, $\xi \in \partial X$, and let $\mathcal{H}_{\xi,p}$ be a hyperplane in $X$. Then the (signed) distance $\bar{d}(x, \mathcal{H}_{\xi,p})$ between a point $x \in X$ and hyperplane $\mathcal{H}_{\xi,p}$ is defined as*

$$\bar{d}(x, \mathcal{H}_{\xi,p}) = d(x,p)\cos\angle_x(p,\xi).$$

Deriving a closed form of the point-to-hyperplane distance for applications from Definition 3.10 is not trivial. However, one can obtain an upper bound of this distance which is given in Proposition 3.11 (see Appendix 4.4 for its proof).

**Proposition 3.11.** *Let $x, p \in X$, $\xi \in \partial X$, and let $a_\xi \in \Delta$ be such that $\xi$ is the equivalent class of the geodesic ray $\delta(t) = k\exp(ta_\xi)K$, where $k \in K$ and $\exp(\cdot)$ is the matrix exponential. Then*

$$\bar{d}(x, \mathcal{H}_{\xi,p}) \leq \langle d_\Delta(x,p), a_\xi \rangle.$$

Note that the point-to-hyperplane distance in Section 3.1 as well as those in [16, 33, 34] are obtained by solving an optimization problem in Euclidean spaces. This is different from our method in this section which estimates an upper bound of the point-to-hyperplane distance on the target spaces.

### 3.3 Neural Networks on Siegel Spaces

In this section, we show how to construct MLR layers for Siegel neural networks using the tools introduced in Sections 3.1 and 3.2. We also propose two types of FC layers which are crucial building blocks in the context of deep neural networks.

#### 3.3.1 MLR Layers

We follow the approach in [16, 26] for building Riemannian MLR. Given $M$ classes, (Euclidean) MLR computes the probability of each of the output classes as

$$p(y = j|x) = \frac{\exp(a_j^T x - b_j)}{\sum_{j=1}^{M} \exp(a_j^T x - b_j)} \propto \exp(a_j^T x - b_j), \tag{3}$$

where $x$ is an input sample, $b_j \in \mathbb{R}$, $x, a_j \in \mathbb{R}^m, j = 1, \ldots, M$, and $\exp(\cdot)$ is the ordinary exponential function (by abuse of notation). As shown in [26], Eq. (3) can be rewritten as

$$p(y = j|x) \propto \exp(\mathrm{sign}(a_j^T x - b_j) \|a_j\| \bar{d}(x, \mathcal{H}_{a_j, b_j}^E)),$$

where $\bar{d}(x, \mathcal{H}_{a_j, b_j}^E)$ is the distance from point $x$ to hyperplane $\mathcal{H}_{a_j, b_j}^E$ (see Section 3.1.1). In our case, a hyperplane can be parameterized by two elements in $X$ (see Definition 3.4), or by an element in $X$ and an element in $\Delta$ (see Definition 3.9). We replace the expression in the argument of the function $\exp(\cdot)$ by the distances in Theorems 3.5 and 3.8 as well as the upper bound of the point-to-hyperplane distance in Proposition 3.11. The final formulations of our MLR layers are given in Appendix 1.

#### 3.3.2 FC Layers

**The FC layer with group action (AFC)**   Let $a + ib \in \mathbb{SH}_m$. Then the element in $\mathrm{Sp}_{2m}$ mapping $iI_m$ to $a + ib$ via the group action (see Section 2.1) is given by

$$\phi(a + ib) = \begin{bmatrix} b^{\frac{1}{2}} & ab^{-\frac{1}{2}} \\ \mathbf{0} & b^{-\frac{1}{2}} \end{bmatrix}.$$

Given an input $x \in \mathbb{SH}_m$, the output of the AFC layer is obtained by taking the group action $\phi(a + ib)[x]$. This leads us to the following construction.

**Definition 3.12.** *Let $x = u + iv \in \mathbb{SH}_m$ be the input of the AFC layer. Then the output of the AFC layer is given by:*

$$t = (b^{\frac{1}{2}} u b^{\frac{1}{2}} + a) + ib^{\frac{1}{2}} v b^{\frac{1}{2}},$$

*where $a \in \mathrm{Sym}_m$ and $b \in \mathrm{Sym}_m^+$ are the parameters of the layer.*

We have that $b^{\frac{1}{2}} u b^{\frac{1}{2}} + a \in \mathrm{Sym}_m$ and $b^{\frac{1}{2}} v b^{\frac{1}{2}} \in \mathrm{Sym}_m^+$ by construction. Hence, the AFC layer always outputs points on $\mathbb{SH}_m$. The transformation performed by the AFC layer can be interpreted as a translation of the input $x$ by $a + ib$ (see Section 3.1.1).

**The FC layer for dimensionality reduction (DFC)**   Based on the definition of the AFC layer, another type of FC layers for Siegel neural networks can also be built using a method similar to [20].

**Definition 3.13.** *Let $\mathrm{St}_{m,m_2}$ be the space of $m \times m_2$ real matrices ($m > m_2$) with mutually orthogonal columns of unit length (the compact Stiefel manifold), and let $x = u + iv \in \mathbb{SH}_m$ be the input of the DFC layer. Then the output of the DFC layer is given by:*

$$t = (b^T u b + a) + ib^T v b,$$

*where $a \in \mathrm{Sym}_{m_2}$ and $b \in \mathrm{St}_{m,m_2}$ are the parameters of the layer.*

Our FC layers generalize some FC layers in previous works. Specifically, when $u = 0$ and $a = 0$, the imaginary part $b^{\frac{1}{2}} v b^{\frac{1}{2}}$ of the output of the AFC layer corresponds to the transformation performed by the affine-invariant translation layer [33], and the imaginary part $b^T v b$ of the output of the DFC layer corresponds to the transformation performed by the well-known Bimap layer [20]. In [40], the authors also proposed FC layers for neural networks on RSS. However, these layers are different from our FC layers in some aspects. First, the former include activation functions which are not used in the latter. Second, the former do not output points on the considered spaces, as opposed to the latter which always output points on these spaces.

| Method | Dataset 1 $(3, 600, 3600)$ | Dataset 2 $(4, 100, 2000)$ | Dataset 3 $(5, 80, 1600)$ | Dataset 4 $(6, 50, 500)$ |
|---|---|---|---|---|
| kNN [11] | 76.22±0.0 | 93.00±0.0 | 76.75±0.0 | 73.20±0.0 |
| SPDNet [20] | 63.44±0.11 | 41.50±0.12 | 45.88±0.15 | 66.80±0.04 |
| SPDNetBN [9] | 62.67±0.10 | 45.10±0.08 | 45.75±0.15 | 68.40±0.04 |
| MLR-AI [33] | 65.61±0.15 | 47.40±0.12 | 46.12±0.17 | 67.60±0.04 |
| GyroSpd++ [34] | 62.24±0.16 | 46.20±0.14 | 48.25±0.19 | 67.80±0.08 |
| SiegelNet-DFC-QMLR$_{\mathrm{Sym}_m^+ \times \mathbb{SH}_m^{q-1}}$ (Ours) | 40.78±0.23 | 82.70±0.18 | 74.88±0.21 | 71.20±0.08 |
| SiegelNet-AFC-QMLR$_{\mathrm{Sym}_m^+ \times \mathbb{SH}_m^{q-1}}$ (Ours) | **80.94±0.14** | **96.50±0.12** | **91.00±0.18** | **85.60±0.06** |

Table 1: Results (mean accuracy ± standard deviation) computed over 10 runs for radar clutter classification. The tuple $(m, M, s)$ below each dataset indicates the signal dimension $m$, the number of classes $M$, and the size of the dataset $s$.

## 4 Related Work

Existing MLR models on Riemannian manifolds are generally built on either SPD manifolds [13, 33] and their low-rank counterparts [34] or hyperbolic spaces [5, 16, 26, 38]. Many of them [16, 33, 34, 38] leverage the gyro-structures of the Poincaré ball and SPD manifolds. The work in [35] proposes MLR and FC layers for neural network on RSS which rely on the construction of Busemann functions. The work in [40] analyzes some existing hyperbolic and SPD neural networks from the perspective of harmonic analysis on RSS. It mainly concerns with a constructive proof of the universal approximation property of finite neural networks on RSS. Our method in Section 3.1 is inspired by the works in [16, 33, 34] and focuses on Siegel spaces. Our method in Sections 3.2 explores the connection between the point-to-hyperplane distance and the vector-valued distance which has not been investigated in previous works.

## 5 Experiments

This section reports results of our experiments on the radar clutter classification and node classification tasks. For further details, please refer to Appendix 1 in which we present more experimental results on human action recognition and Riemannian generative modeling.

### 5.1 Radar Clutter Classification

Radar clutter classification aims at recognizing different types of radar clutter which is the information recorded by a radar related to seas, forests, fields, cities and other environmental elements surrounding the radar [10]. Due to the scarcity of publicly available radar datasets for the task, our experiments are performed using simulated radar signals[1] which are commonly assumed to be stationary centered autoregressive (AR) Gaussian time series [3, 4, 6, 10]. The AR model is given by

$$u_n + \sum_{j=1}^{q} c_j u_{n-j} = v_n,$$

where $q$ ($q > 1$) is the order of the AR model, $u_n \in \mathbb{C}^m$ is the vector of signals at time $n$, $c_j \in \mathbb{C}^{m \times m}$, $j = 1, \ldots, q$ are the prediction coefficients (AR parameters), and $v_n \in \mathbb{C}^m$ is the prediction error at time $n$ which is assumed to be a multidimensional Gaussian random variable (detailed descriptions of the construction of our datasets are provided in Appendix 1.1). To compute an input data for our networks from a time series, we parameterize the time series as $(p_0, w_1, \ldots, w_{q-1}) \in \mathbb{H}_m^+ \times \mathbb{SD}_m^{q-1}$, where $p_0 \in \mathbb{H}_m^+$ and $w_1, \ldots, w_{q-1} \in \mathbb{SD}_m$ (see Appendix 1.1). We note that methods dealing with data that lie on these product spaces have already been studied in previous works [4, 10, 11]. These representation spaces are endowed with a natural metric inspired by information geometry [4, 10]. We discard the imaginary part of the component $p_0$ and map it to an SPD matrix $\tilde{p}_0$ (see Appendix 1.1). Each component $w_i$ is converted to $z_i \in \mathbb{SH}_m$ using the inverse matrix Cayley transformation (see Section 2.1). The input data is thus represented by point $(\tilde{p}_0, z_1, \ldots, z_{q-1}) \in \mathrm{Sym}_m^+ \times \mathbb{SH}_m^{q-1}$.

---

[1] https://github.com/nguyenxuanson10/synthetic-data

| Method | Glass | Iris | Zoo |
|---|---|---|---|
| kNN [11] | 29.65±0.0 | 31.66±0.0 | 33.33±0.0 |
| LogEig classifier [27] | 41.54±4.22 | 34.33±3.46 | 51.04±3.53 |
| SiegelNet-BFC-BMLR [35] | 41.12±3.86 | 37.26±2.53 | 48.12±3.08 |
| SiegelNet-AFC-VMLR (Ours) | 42.06±4.23 | 36.94±3.68 | 50.86±3.26 |
| SiegelNet-AFC-QMLR$_{\mathbb{SH}_m}$ (Ours) | **45.79±4.66** | **38.20±3.03** | **53.37±4.23** |

Table 2: Results (mean accuracy ± standard deviation) computed over 10 runs for node classification.

| Method | Glass | Iris | Zoo |
|---|---|---|---|
| SiegelNet-BMLR [35] | 40.55±3.50 | 36.94±2.09 | 46.43±3.64 |
| SiegelNet-VMLR (Ours) | 41.78±4.11 | 36.89±3.73 | 50.38±3.47 |
| SiegelNet-QMLR$_{\mathbb{SH}_m}$ (Ours) | **42.61±3.26** | **37.52±2.54** | **52.00±4.78** |

Table 3: Comparison (mean accuracy ± standard deviation) of MLR models on Siegel spaces.

Each of our networks consists of an FC (AFC or DFC) layer and a MLR layer built on the distance in Theorem 3.8. The sizes of the parameter $b$ in the DFC layer are set to $3 \times 2$, $4 \times 3$, $5 \times 3$, and $6 \times 4$ for the experiments on datasets 1, 2, 3, and 4, respectively. We compare our approach to the following methods: (1) k-Nearest Neighbors (kNN) based on the Kähler distance [11] which is among the very few works for supervised classification in the product space $\mathbb{H}_m^+ \times \mathbb{SD}_m^{q-1}$; and (2) state-of-the-art SPD neural networks [9, 20, 33, 34] which use the real parts of the covariance matrices estimated from the time series as input data (the real parts are mapped to SPD matrices as above). We use default settings for SPD models as in the original papers (see Appendix 1.1). Results in Tab. 1 show that SiegelNet-AFC-QMLR$_{\text{Sym}_m^+ \times \mathbb{SH}_m^{q-1}}$ yields the best performance in terms of mean accuracy across all the datasets. It is able to improve upon kNN, the second best method, by a margin of 4.71%, 3.5%, 14.25%, and 12.39% on datasets 1, 2, 3, and 4, respectively. There are large gaps in the performance of our models, yet in most cases, our worst model still outperforms SPD models by large margins. The results of our networks and kNN demonstrate the representation power of Siegel spaces in the considered application. This is also confirmed by our experiments (see Appendix 1.1) in which the performance of SiegelNet-AFC-QMLR$_{\text{Sym}_m^+ \times \mathbb{SH}_m^{q-1}}$ drops drastically when the coordinates associated with the product space $\mathbb{SH}_m^{q-1}$ (i.e., $z_1, \ldots, z_{q-1}$) are removed from the input data.

## 5.2  Node Classification

We perform node classification experiments on Glass, Iris, and Zoo datasets from the UCI Machine Learning Repository [14][2]. Like [27], our main aim is to demonstrate the applicability of our approach on Siegel spaces, and we do not necessarily seek state-of-the-art results for the target task.

To create input data which are graph node embeddings on Siegel spaces, we optimize a distance-based loss function [17, 27]. Given the distances $\{d_G(j_1, j_2)\}_{j_1, j_2=1}^M$ between all pairs of connected nodes $j_1$ and $j_2$, the loss function is given by:

$$\mathcal{L}(x) = \sum_{j_1, j_2=1}^M \left| \left( \frac{d_{\mathbb{SH}}(x_{j_1}, x_{j_2})}{d_G(j_1, j_2)} \right)^2 - 1 \right|,$$

where $x_{j_1}$ and $x_{j_2}$ are the node representations on the embedding space of nodes $j_1$ and $j_2$, respectively, and $d_{\mathbb{SH}}(\cdot, \cdot)$ is the distance function given in Section 2.1. This loss function captures the average distortion. We use the cosine distance to compute a complete input distance graph from the original features of the data points [12, 27]. After the node embeddings[3] are learned, they are used as input features for all methods. In our experiments, the embedding dimension is set to 6.

Each of our networks consists of an AFC layer and a MLR (QMLR$_{\mathbb{SH}_m}$ or VMLR) layer. The QMLR$_{\mathbb{SH}_m}$ and VMLR layers are built using the distances in Theorem 3.5 and the upper bound of $\tilde{d}(x, \mathcal{H}_{\xi, p})$ in Proposition 3.11, respectively. We compare our networks to the following methods:

---

[2]https://archive.ics.uci.edu/datasets
[3]https://github.com/nguyenxuanson10/synthetic-data

(1) kNN based on the distance function $d_{\mathbb{SH}}(\cdot, \cdot)$; (2) LogEig classifier [27]; and (3) SiegelNet-BFC-BMLR which consists of an FC (BFC) layer and a MLR (BMLR) layer based on Busemann functions [35]. Results in Tab. 2 show that SiegelNet-AFC-QMLR$_{\mathbb{SH}_m}$ gives the best mean accuracies across all the datasets. In terms of mean accuracy, SiegelNet-AFC-VMLR surpasses SiegelNet-BFC-BMLR on Glass and Zoo datasets. SiegelNet-AFC-VMLR also surpasses the LogEig classifier on Glass and Iris datasets. Tab. 3 reports the results of SiegelNet-BFC-BMLR and our networks without FC layers. It can be observed that our MLR models achieve higher (mean) accuracies than the BMLR model. Specifically, SiegelNet-QMLR$_{\mathbb{SH}_m}$ improves upon SiegelNet-BMLR by a margin of 2.06%, 0.58%, and 5.57% on Glass, Iris, and Zoo datasets, respectively. SiegelNet-AFC-QMLR$_{\mathbb{SH}_m}$ is able to improve by 3.17%, 0.67%, and 1.36% w.r.t. SiegelNet-QMLR$_{\mathbb{SH}_m}$ on Glass, Iris, and Zoo datasets, respectively, demonstrating the effectiveness of the AFC layer. Although SiegelNet-VMLR is outperformed by SiegelNet-QMLR$_{\mathbb{SH}_m}$, it is important to note that the model size of the former is about two times smaller than that of the latter (see Appendix 1.2).

# 6 Limitation of Our Approach

A limitation of our method in Section 3.1 is that our formulation of Siegel hyperplanes suffers from an over-parameterization issue. We alleviate this problem by reparameterizing Siegel hyperplanes as proposed in Section 3.2. However, this new parameterization does not yield competitive performance compared to the original one.

Our methods rely on operations on Siegel spaces which are generally expensive. Our method in Section 5.1 suffers from high computational cost in the setting of high-dimensional radar signals. Similarly, the loss function in Section 5.2 is based on the average distortion, for which the distances over all pairs of points must be computed during training. Since the computation of the Riemannian distance between two points on a Siegel space (see Section 2.1) is based on eigenvalue decomposition, our method in Section 5.2 is computationally expensive when it comes to learning on large graphs.

Like hyperbolic and SPD spaces, Siegel spaces are spaces of non-positive curvature. Therefore, our method in Section 5.2 does not allow isometric embeddings of graphs with a different curvature property, e.g., non-negative curvature. Although it can still be applied in this case, the learned node embedding may not preserve the curvature property of the embedded graph, leading to poor performance. Furthermore, like other graph embedding approaches, low-dimensional embeddings on Siegel spaces are not able to capture complex relationships within data which can affect the performance of our method.

# 7 Conclusion

We have proposed Riemannian MLR and FC layers which enable the construction of effective Siegel neural networks. Our MLR layers are built upon the quotient structure of Siegel spaces and the concept of vector-valued distance on RSS. Our FC layers are based on the action of the real symplectic group on Siegel spaces. We have provided experimental evaluations demonstrating state-of-the-art performance of our approach in the radar clutter classification and node classification tasks.

There are several potential improvements and extensions to Siegel neural networks that could be addressed as future work. Based on our experimental results, it can be observed that the DFC layer gives inferior performance compared to the AFC layer. It is therefore desirable to develop alternative layers for the DFC layer which are able to achieve better performance. Also, important building blocks such as convolutional layers, batch normalization layers, pooling layers, and attention layers are not studied in our work. Those are crucial to the development of effective deep Siegel neural networks.

# Acknowledgments

We are grateful for the constructive comments and feedback from the anonymous reviewers.

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
