# OpenReview forum: "Siegel Neural Networks"
_NeurIPS.cc/2025/Conference — NeurIPS 2025 poster_

### Official Review · Reviewer_qfip · 2025-06-21

**Clarity:** 3
**Significance:** 3
**Originality:** 3
**Rating:** 5
**Confidence:** 3

**Summary:**

This work focuses on designing neural networks adapted to Siegel spaces. It introduces two type of layer: 1) fully connected layers on the Siegel upper half space, and 2) multiclass logistic regression layers. The second type of layers rely on defining a suitable notion of hyperplane and of distance to the hyperplane. The authors propose two ways to extend hyperplanes to Riemannian symmetric space of non compact type, and in particular on Siegel spaces. First, they use gyrovectors operations, and show that they the distance to the hyperplane has a closed-form in this case. Second, they also propose to use vector values distances. Finally, they show on Radar clutter classification and node classification tasks the potential of their architecture to deal with data living on Siegel spaces.

**Questions:**

In Definition 3.12, it is stated that the distance between $x$ and the hyperplane is defined as $\bar{d}(x, \mathcal{H}_{\xi,p})=\langle d_\Delta(x,p), a_\xi\rangle$. But in my understanding, it is only an upper bound, which is thus confusing. Mayber you could change the notation?

In Table 1, do you have an explanation why the AFC layers are always better than the DFC layers. Also, why aren't the vector valued layers compared on the radar data?

Comparisons in Table 3 are also done with Busemann layers. I guess it requires to compute the Busemann function. How is it computed on Siegel spaces? Why isn't it used in the experiment of Table 1?

There is an analysis of the complexity in Appendix 1. But it is not clear how it compares with the baselines used in the experiments sucha as the Busemann layers?

**Ethical Concerns:**

["NO or VERY MINOR ethics concerns only"]

**Final Justification:**

This work introduce the first neural networks on Siegel spaces. I believe that it is a very interesting contribution, as Siegel spaces are known to be more expressive than hyperbolic spaces or SPD spaces to embed complex data.

Some concerns were raised about the mathematical contribution and on the applications of such networks. But the authors convinced me of the importance of their contributions. Thus, I recommend acceptance.

**Limitations:**

yes

**Quality:**

4

**Strengths And Weaknesses:**

Few works have focused on Siegel spaces in Machine Learning. Therefore, it is nice to see a work focusing on defining tractable neural networks on this space. The proposed ways to extend notions of hyperplanes to Siegel spaces are very interesting, and pretty clear to understand. This results in layers that can be used to define neural networks, which are shown to work pretty well on real data, which is a very nice contribution.

**Strengths**:
- Derivation of 2 extension of hyperplanes on Riemannian symmetric spaces along with distances to hyperplanes
- Two type of layers to define neural networks: MLR and fully connected
- Application on real data, showing improvements over baselines

**Weaknesses**:
- The presentation of gyrovectors results is close to the one of [1], and it seems that the main contribution here is its specification to Siegel spaces.
- The comparison between the proposed layers could be a bit improved.
- For the vector-values distance, the distance to hyperplane is not known in closed-form.

[1] Nguyen, X. S., Yang, S., & Histace, A. (2025). Neural networks on Symmetric Spaces of Noncompact Type. In The Thirteenth International Conference on Learning Representations.

---

> ### Author Rebuttal · Authors · 2025-07-31
>
> We thank the reviewer for their insightful comments and questions.
>
> *The presentation of gyrovectors results is close to the one of [1], and it seems that the main contribution here is its specification to Siegel spaces.*
>
> Our paper presents three new technical contributions with respect to [33] (the reference given by the reviewer). First, our method in Section 3.1 relies on a gyrovector space approach which is different from the approach in [33] based on Busemann functions. This allows us to derive a closed-form for the point-to-hyperplane distance in the setting of product spaces, which is not trivial for the approach in [33]. Second, our method in Section 3.2 is built upon vector-valued distances which do not appear in [33]. Third, the construction of our FC layers is fundamentally different from that of FC layers in [33] which is based on the point-to-hyperplane distance. From the practical aspect, to the best of our knowledge, we build the first Siegel neural networks for solving the tasks of radar clutter classification and action recognition (see Appendix 1.1.4).
>
> *In Definition 3.12, it is stated that the distance between $x$ and the hyperplane is defined as $\bar{d}(x, \mathcal{H}\_{\xi,p})=\langle d\_{\Delta}(x,p), a\_{\xi}\rangle$. But in my understanding, it is only an upper bound, which is thus confusing. Mayber you could change the notation?*
>
> To avoid confusion, we propose to denote the upper bound of $\bar{d}(x, \mathcal{H}\_{\xi,p})$ by $\tilde{d}(x, \mathcal{H}\_{\xi,p})$ which is defined as $\tilde{d}(x, \mathcal{H}\_{\xi,p})=\langle d\_{\Delta}(x,p), a\_{\xi}\rangle$. We can then remove Definition 3.12 and in the experiments, we will state that the QMLR layer is built upon the upper bound of $\bar{d}(x, \mathcal{H}\_{\xi,p})$.
>
> *In Table 1, do you have an explanation why the AFC layers are always better than the DFC layers. Also, why aren't the vector valued layers compared on the radar data?*
>
> We agree that the answer to this question would give one some insight on how to design better FC layers for dimensionality reduction on Siegel spaces. However, we do not think there is an obvious answer. A potential reason could be that learning our network parameters involves solving a highly nonlinear optimization problem, and all the parameters of the AFC layer are optimized on Euclidean spaces, while the parameter $b$ of the DFC layer is learned using Riemannian optimization on Stiefel manifolds.
>
> Regarding the radar clutter classification experiments, the input data lie on products of SPD and Siegel spaces. Our proposed VMLR layer is designed on Siegel spaces and cannot deal with data on product spaces (it is not trivial to adapt them to this setting using a similar derivation in Section 3.2, so we did not consider investigating this idea in our paper). For this reason, we did not use them in these experiments.
>
> *Comparisons in Table 3 are also done with Busemann layers. I guess it requires to compute the Busemann function. How is it computed on Siegel spaces? Why isn't it used in the experiment of Table 1?*
>
> The key point in the computation of the distance [33] based on Busemann functions is the computation of the map $H: G \rightarrow \mathfrak{a}$ determined by $g_1 = k\_1 \exp H(g\_1) n\_1$ with $g\_1 \in G$, $k\_1 \in K$, $n\_1 \in N$, and $\mathfrak{a}$ is the Lie algebra of $A$ (please refer to Proposition 4.9 and Corollary 4.10 in [33]). We recall here that any connected noncompact semisimple Lie group with finite center $G$ can be decomposed into subgroups $K$, $A$, and $N$, where $K$ is maximal compact, $A$ is maximal abelian, and $N$ is maximal nilpotent (the decomposition in question is called Iwasawa decomposition). The map $H(\cdot)$ can be computed by Cholesky factorization [A9, A10] as follows. Let $g = \begin{bmatrix} a & b \\\\ c & d  \end{bmatrix} \in \operatorname{Sp}_{2m}$ and let $g^Tg = \begin{bmatrix} a\_1 & b\_1 \\\\ b\_1^T & d\_1 \end{bmatrix}$.
> Let $a\_1 = q^Trq$ be the Cholesky factorization of the positive definite matrix $a_1$, where $q$ is a unit upper triangular matrix and $r$ is a diagonal matrix with positive diagonal entries. Then $H(g) = \begin{bmatrix} \frac{1}{2}\log(r) & 0 \\\\ 0 & -\frac{1}{2}\log(r) \end{bmatrix}$. We propose to add these details to Appendix 1.2.2.
>
> For the same reason regarding the VMLR layer, the BMLR layer is not designed on product spaces and therefore is not evaluated for
> radar clutter classification experiments.
>
> *There is an analysis of the complexity in Appendix 1. But it is not clear how it compares with the baselines used in the experiments sucha as the Busemann layers?*
>
> The BMLR layer has the same memory and time complexities as the VMLR layer, i.e., their memory complexity is $O(Mm(m+2))$ and
> their time complexity is $O(Mm^3)$. We propose to add this analysis to Appendix 1.2.3.
>
> **References**
>
> [A9] P. Sawyer. Computing the Iwasawa Decomposition of the Classical Lie Groups of Noncompact Type Using the QR Decomposition. Linear Algebra and its Applications, 493:573–579, 2016.
>
> [A10] T.-Y. Tam, Computing the Iwasawa decomposition of a symplectic matrix by Cholesky factorization, Appl. Math. Lett. 19 (12) (2006) 1421–1424.

---

> > ### Comment · Reviewer_qfip · 2025-08-04
> >
> > Thank you for answering my questions and my concerns. I am fully convinced by the answers and the additional experiments provided, and I will raise my score.

---

> > > ### Author Response · Authors · 2025-08-08
> > >
> > > We are glad that our answers have addressed your concerns. Thank you again for your thorough review and insightful questions.

---

### Official Review · Reviewer_ahAK · 2025-06-25

**Clarity:** 3
**Significance:** 3
**Originality:** 4
**Rating:** 4
**Confidence:** 3

**Summary:**

This paper introduces Siegal Neural Networks, meant to function on the dataset of underlying siegal space structure, which is roughly a tensorial generalization of Hyperbolic space.  The paper establishes the analogue of FC layer(AFC, DFC) and classification readout layer(MLR) by closely aligning (1) the linear transformation in usual FC to the group transformation of Sp2m, (2) “distance to plane” component of the softmax to the “distance to the hyperplane” in SHm.
The paper establishes closed form as well as the surrogate for these “distance” functions.
The paper provides empirical results showcasing the efficacy of the Neural network consisting of XFC and MLR.

**Questions:**

What is the true advantage of Weyl chamber based definition (3.12?). The mention is being made about the reduction of the parameter size, but it does neither changes the “order-level” cardinality of the parameter set  nor play out too well in the final performance.    Is it also a legitimate “distance”?

In regards to the point made in the weakness section, is there not a dataset/task  that “only” accepts computations closed in SHm?  Are there any intuitive cases where the use of Siegel Networks would be natural?

**Ethical Concerns:**

["NO or VERY MINOR ethics concerns only"]

**Final Justification:**

I am suggesting the acceptance with score 4.  I could not raise the score to 5 because I was unable to obtain the clear answer  to the question of “Why Siegel? Why not Hyperbolic?” has not been fully answered.  The question was answered partially through its reference to [28], but the experiments were not conducted on the tasks discussed in [28], by which the research may be able to further strengthen its evidence of utility. At the same time, I still maintain my praise for the research in regards to its theoretical and fundamental contributions, as well as its provision of a new venue of framework.

**Limitations:**

Limitation is being mentioned in Appendix Section, with regard to the overparameterization.    I wonder if there is also possible limitation with respect to the representation power, caused by the structural inductive bias to the network.

**Quality:**

2

**Strengths And Weaknesses:**

Strength:

The paper introduces mathematically principled extension of the usual MLP to Siegel space by theoretically assuring each intermediate output to live in SHm and by aligning the point-to-plane distance component of the softmax to the Siegel Geometry.


The paper provides closed forms and analogues and upper bounds that seems useful not only for the construction of their propose network but also to other applications on Siegel Space


Weaknesses

Some parts of the manuscript were simply hard to follow. For example, I presume that, in 3.2.2, the definition of dbar changes over the paragraph— starting from Def 3.10, it leverages the prop 3.11 to “redefine” dbar in 3.12.   This was very confusing.  The motivation of 3.2 was also not strong enough for me— it took a while until I realized that this section was dedicated for the proposal “approximate surrogate” as opposed to a strictly better version of 3.5.  Also,  Eq(1) referred wrongly in the intro of 3.2?  I believe the authors wanted to refer to definition 3.4?   It also confused me that QMLR appeared in 5.1 (it is first described in 5.2)



While I am hesitant in evaluating the paper of theoretical significance as this research based on the quality of experiments, I am a little worried about the implications of the results— although the construction of Siegel Networks on their own is intellectually exciting and promising,  I am afraid “Why Siegel? Why not Hyperbolic?” is not fully justified in the end.  With such interesting results regarding Siegel Space, I believe the paper is obliged to demonstrate to the community the clear “practical” symmetric difference of Siegel space based approach from Hyperbolic space approach.
For example, it is being claimed in section 5.1 that  "to the best of our knowledge , there is no baseline in our context to compare against";  however, because the task on its own is a classification problem, I find it hard to believe that the dataset cannot be embedded to other spaces, such as Poincare Ball, Euclidean Space,  etc.  In such a case, Hyperbolic Networks and other conventional methods  would be the true adversary.  I wonder if the same applies to Node classification. 　For this ground I feel that the statements in abstract regarding the experimental performance is overstated.
While I completely agree that the focus of this paper should not be the invention of a SOTA method for a popular scientific task,  I feel that it is important to bring about at least one case in which only the Siegel network can solve.

---

> ### Author Rebuttal · Authors · 2025-07-31
>
> We thank the reviewer for their insightful comments and questions.
>
> *Some parts of the manuscript were simply hard to follow. For example, I presume that, in 3.2.2, the definition of dbar changes over the paragraph— starting from Def 3.10, it leverages the prop 3.11 to “redefine” dbar in 3.12. This was very confusing.*
>
> To avoid confusion, we propose to denote the upper bound of $\bar{d}(x, \mathcal{H}\_{\xi,p})$ by $\tilde{d}(x, \mathcal{H}\_{\xi,p})$ which is defined as $\tilde{d}(x, \mathcal{H}\_{\xi,p})=\langle d\_{\Delta}(x,p), a\_{\xi}\rangle$. We can then remove Definition 3.12 and in the experiments, we will state that the QMLR layer is built upon the upper bound of $\bar{d}(x, \mathcal{H}\_{\xi,p})$.
>
> *The motivation of 3.2 was also not strong enough for me— it took a while until I realized that this section was dedicated for the proposal “approximate surrogate” as opposed to a strictly better version of 3.5.*
>
> Please see our answer to the question related to this comment below.
>
> *Also, Eq(1) referred wrongly in the intro of 3.2? I believe the authors wanted to refer to definition 3.4? It also confused me that QMLR appeared in 5.1 (it is first described in 5.2)*
>
> We would like to state that "the formulation in Eq. (1) has an over-parameterization issue, our formulation in Definition 3.4 follows that formulation and thus suffers from a similar issue." We think Eq. (1) is correctly referred, but we agree that there is a missing connection between the first two sentences of Section 3.2. We propose to rephrase them as above to avoid confusion.
>
> The QMLR layer in Section 5.2 (constructed from the distance in Theorem 3.5) is only a special instance of the ProductQMLR layer in Section 5.1 (constructed from the distance in Theorem 3.8). This is because when the Cartesian product contains only one Siegel space, the distance in Theorem 3.8 is precisely the one in Theorem 3.5. To avoid confusion, we propose to call both layers as QMLR and use subscripts to indicate the components of the considered space. For instance, the ProductQMLR layer will be called as QMLR$\_{\operatorname{Sym}\_m^+ \times \mathbb{SH}\_m^L}$, and the QMLR layer will be called as QMLR$\_{\mathbb{SH}_m}$, where $L$ is the number of Siegel spaces in the Cartesian product.
>
> *I believe the paper is obliged to demonstrate to the community the clear “practical” symmetric difference of Siegel space based approach from Hyperbolic space approach.*
>
> The practical advantage of Siegel spaces compared to Euclidean spaces, hyperbolic spaces, products of Euclidean and hyperbolic spaces, and SPD spaces has been demonstrated in [28] in the context of graph embedding for a variety of applications (e.g., graph reconstruction, recommender systems, analysis of the embedding space). It is observed from their experimental results that the use of Siegel spaces in combination with appropriate metrics generally results in the best performance. Please refer to [28] for more details.
>
> In our work, the practical advantage of Siegel spaces compared to hyperbolic spaces can be seen in radar clutter classification in which the use of Siegel networks is natural and results in the best performance, whereas the use of hyperbolic networks is not trivial. We have also shown that the task of action recognition can be tackled by Siegel networks as opposed to hyperbolic networks which are not commonly adopted for solving such a problem. Please refer to our response to Reviewer 8RJ3 for more experimental results.
>
> *For example, it is being claimed in section 5.1 that "to the best of our knowledge , there is no baseline in our context to compare against"; however, because the task on its own is a classification problem, I find it hard to believe that the dataset cannot be embedded to other spaces, such as Poincare Ball, Euclidean Space. I wonder if the same applies to Node classification.*
>
> One can use conventional models for classification of time series, e.g., RNN and LSTM in hyperbolic and Euclidean spaces to tackle the problem in Section 5.1. However, these models were originally designed to work with real-valued data, and it is not clear how effective they are for complex-valued data. In our experiments, we tested an LSTM for complex-valued data using the ComplexNN library implemented in Pytorch (we cannot provide its link due to the policy of NeurIPS 2025). However, we obtained very poor results on our datasets.
>
> For the task of node classification, the performance of our method depends not only on the proposed FC and MLR layers but also on the method used for learning node embeddings. Since the focus of our work is to build neural networks on RSS, we think it would be more important to validate these layers by comparing them with existing ones on Siegel spaces.
>
> Since there is possibility to use methods in hyperbolic and Euclidean spaces (or even other Riemannian manifolds) for comparison in Section 5.1 (although it is not guaranteed that they work well for our problem), we propose to remove the statement "to the best of our knowledge, ..." in this section.
>
> *What is the true advantage of Weyl chamber based definition (3.12?)... Is it also a legitimate “distance”?*
>
> From a theoretical aspect, the distance in Definition 3.12 has a nice property, i.e., it is $K$-invariant. This can be seen by first noting that $d\_{\Delta}(x,p) = d\_{\Delta}(k\_1[x],k\_1[p])$ for any $k\_1 \in K$ due to the $G$-invariant property of the vector-valued distance $d\_{\Delta}(\cdot,\cdot)$, and $a\_{\xi} = a\_{k\_1[\xi]}$ since $k\_1[\xi]$ is the equivalent class of the geodesic ray $k\_1[\delta(t)] = k\_1k\exp(ta\_{\xi})K = k\_2\exp(ta\_{\xi})K$ where $k\_2 \in K$ (see the notations in Proposition 3.11). Therefore, $\bar{d}(x,\mathcal{H}\_{\xi,p}) = \left\langle d\_{\Delta}(x,p),a\_{\xi} \right\rangle = \left\langle d\_{\Delta}(k\_1[x],k\_1[p]),a\_{k\_1[\xi]} \right\rangle = \bar{d}(k\_1[x],\mathcal{H}\_{k\_1[\xi],k\_1[p]})$. It is easy to see that this property does not hold for the distance in Section 3.1.
>
> The distance in Definition 3.12 when applied to Siegel spaces results in the VMLR layer which has the same memory and time complexities as the BMLR layer (please refer to our response to the last comment of Reviewer qfip). The former improves upon the latter on node classification in most cases (see Table 3). Furthermore, our experiments on Riemannian generative modeling demonstrate the effectiveness of this distance in such an application. Please refer to our response to Reviewer 8RJ3 for details on our experiments.
>
> Since the distance in Definition 3.12 is not the shortest distance between a given point and any point on a given Siegel hyperplane,
> it is not a legitimate point-to-hyperplane distance in this sense.
>
> *It is important to bring about at least one case in which only the Siegel network can solve. In regards to the point made in the weakness section, is there not a dataset/task that “only” accepts computations closed in SHm? Are there any intuitive cases where the use of Siegel Networks would be natural?*
>
> To our knowledge, we are not aware of any dataset/task in which only input data/features on Siegel spaces are available. In our work, the use of Siegel networks for radar clutter classification is well-motivated. Algorithm 1 (see Appendix 1.1.2) is a robust method for estimating autocorrelation matrices which have proven effective in signal processing applications. This motivates its usage for computing the input data of our networks. Due to the assumption that radar signals are stationary centered autoregressive (AR) Gaussian time series, it has been shown in [10] (see Section 2.2.9, Theorem 3) that the output of Algorithm 1 belong to product space $\mathbb{H}\_m^+ \times \mathbb{SD}\_m^{q-1}$. Therefore, the use of Siegel networks comes naturally from the effectiveness of autocorrelation matrices in capturing patterns within data and from the above assumption on radar signals. It has been shown in previous works [21, 22, 23] that different representation spaces (i.e., SPD manifolds, Lie groups, Grassmann manifolds) of the same input data can lead to enhanced performance using Riemannian neural networks.
>
> We note that methods dealing with data that lie on product spaces $\mathbb{H}\_m^+ \times \mathbb{SD}\_m^{q-1}$ have already been studied in previous works [4, 10, 11]. These representation spaces are endowed with a natural metric inspired by information geometry [4, 10].
>
> *I wonder if there is also possible limitation with respect to the representation power, caused by the structural inductive bias to the network.*
>
> Like hyperbolic and SPD spaces, Siegel spaces are spaces of non-positive curvature. Therefore, our method in Section 5.2 does not allow isometric embeddings of graphs with a different curvature property, e.g., non-negative curvature. Although it can still be applied in this case, the learned node embedding may not preserve the curvature property of the embedded graph, leading to poor performance for node classification.
>
> Furthermore, like other graph embedding approaches, low-dimensional embeddings on Siegel spaces are not able to capture complex relationships within data which can affect the performance of node classification.
>
> In addition to the above limitations with respect to the representation power, our methods rely on operations on Siegel spaces which are generally expensive. The loss function in Section 5.2 is based on the average distortion. That is, the distances over all pairs of points must be computed during training. Given that the computation of the Riemannian distance between two points on a Siegel space (see Section 2.1) is based on eigenvalue decomposition, our method is computationally expensive when it comes to learning on large graphs. Similarly, our method in Section 5.1 suffers from high computational cost in the setting of high-dimensional radar signals.

---

> > ### Comment · Reviewer_ahAK · 2025-08-04
> > **Thank you very much for the rebuttal**
> >
> > Thank you very much for the detailed responses, I have acknowledged the suggestions and also the additional experiments.
> > While I believe that additional experiments of emphasizing the utility of the method is still very helpful, I would like to keep my score for now as it is already above 3 and I am inclined to suggest the publication of this work.
> > Also, now that [28] is being re-mentioned as the previous research that addresses the practical advantage of using Siegel Space over other geometry, is there any way to apply the Siegel Neural Networks to the downstream tasks used in [28]? ( I am aware that [28]+LogEigClassifier itself is compared against the proposed method in Node classification.)

---

> > > ### Author Response · Authors · 2025-08-04
> > >
> > > We thank the reviewer for their feedback and an insightful question.
> > >
> > > We think the proposed layers can be used in combination with metric learning to solve the task of recommender systems in [28] (their effectiveness in such an application must be validated by experiments though). For example, once user and item embeddings on Siegel spaces have been learned as in [28], our FC layers can be used to capture interactions between the user and item embeddings for rating prediction [A11]. In Euclidean space, these interactions can be modeled by simply concatenating the user and item embeddings and then feeding the resulting embeddings to an FC layer. In our case, one can average the user and item embeddings by using the binary operation in Definition 3.1 and then feed the resulting embeddings (which lie on Siegel spaces) to the AFC (DFC) layer.
> > >
> > > **References**
> > >
> > > [A11] Shuai Zhang, Lina Yao, Aixin Sun, Yi Tay: Deep Learning Based Recommender System: A Survey and New Perspectives. ACM Comput. Surv. 52(1): 5:1-5:38 (2019).

---

### Official Review · Reviewer_7CnN · 2025-07-03

**Clarity:** 3
**Significance:** 3
**Originality:** 3
**Rating:** 5
**Confidence:** 3

**Summary:**

The paper introduces SiegelNet, the first discriminative neural-network architecture whose layers operate natively on Siegel spaces, a broad family of Riemannian symmetric spaces that strictly generalizes both hyperbolic space and the SPD manifold.
- Two multiclass-logistic-regression (MLR) layers are derived: QMLR, obtained from a new closed-form point-to-hyperplane distance based on the quotient structure of Siegel spaces, and VMLR, a more compact alternative built from vector-valued distance.
- Two fully-connected layers (AFC, DFC) are proposed; DFC also performs dimensionality reduction.
- Combining these ingredients yields SiegelNet, which attains state-of-the-art accuracy on (i) four simulated radar-clutter datasets and (ii) three UCI node-classification benchmarks, consistently outperforming k-NN, SPD and hyperbolic baselines.

**Questions:**

Please refer to the Weaknesses section

**Ethical Concerns:**

["NO or VERY MINOR ethics concerns only"]

**Final Justification:**

As the relation to prior research has been clarified and it is now clearer that this work is novel, I will raise my score.

**Limitations:**

yes

**Quality:**

3

**Strengths And Weaknesses:**

# Strengths
- This study extends the ideas behind gyrovector-based hyperbolic neural networks and neural networks on the SPD manifold to the Siegel domain. Its originality lies in the development of techniques that tackle the new geometric challenges that arise when broadening the class of manifolds up to Siegel domain.

# Weaknesses
- On the other hand, Sonoda et al. (2022) have already applied Helgason’s harmonic analysis to design geometric neural networks on general non-compact symmetric spaces and to show their universality. A discussion clarifying the differences between the present work and that study is therefore requested.
  [SII2022] Sonoda-Ishikawa-Ikeda, Fully-Connected Network on Noncompact Symmetric Space and Ridgelet Transform based on Helgason-Fourier Analysis, ICML2022

---

> ### Author Rebuttal · Authors · 2025-07-31
>
> We thank the reviewer for their insightful comments.
>
> *On the other hand, Sonoda et al. (2022) have already applied Helgason’s harmonic analysis to design geometric neural networks on general non-compact symmetric spaces and to show their universality. A discussion clarifying the differences between the present work and that study is therefore requested.*
>
> The work in [A8] analyzes some existing hyperbolic and SPD neural networks from the perspective of harmonic analysis on RSS. This work mainly focuses on a constructive proof of the universal approximation property of finite neural networks on RSS, and it does not concern with practical applications on RSS.
>
> In terms of network architecture, their considered FC layers are different from our FC layers in some aspects. First, the former include activation functions which are not used in the latter. Second, the former do not output points on the considered spaces, as opposed to the latter which always output points on these spaces.
>
> We propose to add the first paragraph of the above discussion to Section 4 (Related work) and the second one to Section 3.3.2 (FC layers).
>
> **References**
>
> [A8] Sho Sonoda, Isao Ishikawa, and Masahiro Ikeda. Fully-Connected Network on Noncompact Symmetric Space and Ridgelet Transform based on Helgason-Fourier Analysis. In ICML, pp. 20405-20422, 2022.

---

> > ### Comment · Reviewer_7CnN · 2025-08-05
> >
> > Thank you for addressing my questions. I am now convinced by your answers, so I’ll be raising my score.

---

> > > ### Author Response · Authors · 2025-08-08
> > >
> > > We are glad that our answers have addressed your concerns. Thank you again for your thorough review and constructive feedback.

---

### Official Review · Reviewer_8RJ3 · 2025-07-03

**Clarity:** 3
**Significance:** 2
**Originality:** 2
**Rating:** 4
**Confidence:** 3

**Summary:**

The authors build neural networks for Siegel spaces (i.e., space of complex-valued matrices A+ iB such that A is real symmetric and B is real positive-definite symmetric). These spaces generalize the Poincare upper half-plane model of hyperbolic geometry. They propose multiclass logistic regression  (odds) and fully-connected layers for these spaces via gyrovector approach, and the paper is concluded with experiments.

**Questions:**

This is an interesting paper. The authors define distances on Siegel spaces which lead to multi-class odds in Equation 2 for classification in these spaces, as well as linear layers in Section 3.2.2.



### Major Comments/Question:

- The paper is less accessible to the community, as it involves a lot of deep mathematical definitions and hard to understand. Moreover, the theory while being plausible, is not well motivated. My main concern is about the applicability of the results in practice, and the relevance.


### Other Comments:

- Line 72: Can you explain a bit what sort of Riemannian metric induces this distance?

- Line 84: I'm a bit confused; can we obtain any RSS via some G and some K?

- Definition 3.2: How do you define $g^T$ for elements of the group? Are those coming from matrix representations of a Lie group? How do you define log in the same definition?

**Ethical Concerns:**

["NO or VERY MINOR ethics concerns only"]

**Final Justification:**

I had a number of concerns/questions that have been addressed in part by the authors during the discussion period. I decided to update my score from 2 to 4 conditioned on applying the list of promised changes to the paper.

**Limitations:**

yes

**Quality:**

3

**Strengths And Weaknesses:**

Pros:

- interesting problem
- well written


Cons:

- this setting, while being theoretically interesting, is not well motivated in terms of applications

---

> ### Author Rebuttal · Authors · 2025-07-30
>
> We thank the reviewer for their comments.
>
> *Line 72: Can you explain a bit what sort of Riemannian metric induces this distance?*
>
> The distance in line 72 is induced by the following line element [34, 37]:
> \begin{equation*}
> ds^2 = 2 \operatorname{Tr}(v^{-1}dxv^{-1}d\bar{x}), \hspace{3mm} x = u + iv \in \mathbb{SH}_m,
> \end{equation*}
> where $\operatorname{Tr}(\cdot)$ is the matrix trace.
>
> *Line 84: Can we obtain any RSS via some G and some K?*
>
> Please refer to Appendix 3.1 (lines 172 - 176) for details.
>
> *Definition 3.2: How do you define for elements of the group? Are those coming from matrix representations of a Lie group?
> How do you define log in the same definition?*
>
> The space $X$ can be expressed as a quotient of a Lie group. The elements of $X$ are specified from its quotient structure. For Siegel spaces, it is described in Section 2.1. The matrix representation of the Lie group and the map $\phi(\cdot)$ in Section 2.1 are used to give the matrix representation of the elements of $X$.
>
> The function $\log(\cdot)$ denotes the ordinary matrix logarithm.
>
> *The paper is less accessible to the community, as it involves a lot of deep mathematical definitions and hard to understand. Moreover, the theory while being plausible, is not well motivated. My main concern is about the applicability of the results in practice, and the relevance.*
>
> The concepts used in our paper are not new in the context of our work. Section 3.1 uses a gyrovector space approach which has been developed since the pioneering work in [17]. Section 3.2 is based on vector-valued distances which have been studied in [28, 29]. Our work is motivated by the lack of an effective framework for building discriminative Siegel networks. Please refer to Section 1 for our motivation.
>
> Our method in Section 5.1 could result in new tools for analyzing complex stationary centered autoregressive Gaussian time series and Block-Toeplitz matrices [4, 10].
>
> Below we present experiments to further demonstrate the applicability of our method.
>
> ## **Riemannian Generative Modeling**
> To our knowledge, we are not aware of any generative methods that target Siegel spaces. Since we do not aim to propose new methods in this context, we consider SPD spaces and apply the method in [A1] based on flow matching to solve the task. Note that our methods are directly applicable to SPD spaces.
>
> ### **Datasets and Experimental Settings**
> **BCIC-IV-2a [A2]**
> It consists of electroencephalography (EEG) data captured from $9$ subjects. The cue-based BCI paradigm consists of $4$ different motor imagery tasks, namely the imagination of movement of the left hand (class $1$), right hand (class $2$), both feet (class $3$), and tongue (class $4$). Two sessions on different days are recorded for each subject. Each session is comprised of $6$ runs separated by short breaks. One run consists of $48$ trials ($12$ trials for each of the $4$ possible classes), yielding a total of $288$ trials per session.
>
> **MAMEM-SSVEP-II [A3]**
> It consists of EEG data with $256$ channels captured from $11$ subjects executing a SSVEP-based experimental protocol. Five different frequencies ($6.66$, $7.50$, $8.57$, $10.00$ and $12.00$Hz) are used for the visual stimulation, and the EGI $300$ Geodesic EEG System (GES $300$), using a $256$-channel HydroCel Geodesic Sensor Net (HCGSN) and a sampling rate of $250$ Hz is used to capture the signals.
>
> We create two datasets of SPD matrices by computing a covariance matrix from each signal of the above datasets. For BCIC-IV-2a, the session 1 data of subject 1 is used as the training set whose 1/8 is used as the validation set. The session 2 data of subject 1 is used as the test set [A2]. For MAMEM-SSVEP-II, the first 4 sessions of subject 1 are used as the training set whose 1/4 is used as the validation set. The fifth session of subject 1 is used as the test set [A3].
>
> ### **Optimization and Hyperparameters**
> We use the official code of [A1] for our experiments. Vector fields are parameterized as neural networks in the ambient space and are projected onto the tangent space at every point. They are normalized by the inverse of square root of the metric tensor to cancel out the effect of the metric tensor on the Riemannian norm [A1]. Inspired by [A4] which uses a Poincaré MLR layer to construct the vector field, we replace the first convolutional layer of the original vector field (a multilayer perceptron) with the VMLR and QMLR layers, as well as the MLR-LE and MRL-AI layers [31]. The vector-valued distance is computed as in [29]. We use $1000$ Euler steps with a projection after every step for evaluation. We use the checkpoint that gives the best negative log-likelihood (NLL) on the validation set to compute the NLL on the test set [A1]. The numbers of hidden units and layers of the vector field are set to $32$ and $4$, respectively. All networks are trained using AdamW optimizer with model exponential moving average. The learning rate and weight decay are set to $2e-4$ and $0.999$, respectively. The number of epochs and the batch size are set to $100$ and $128$, respectively. Experiments are performed using machines with an Intel Core i7-9700 CPU 3.00 GHz.
>
> | Method | BCIC-IV-2a | MAMEM-SSVEP-II |
> |-----------------------|-------------|-------------|
> |     RFM [A2]       |    -225.23$\pm$3.42    |   -169.28$\pm$1.24      |
> |   RFM-MLR-LE [31]  |   -231.64$\pm$6.19    |    -170.37$\pm$3.18    |
> |   RFM-MLR-AI [31]    |    -238.59$\pm$6.38    |    -170.92$\pm$4.05    |
> |   RFM-QMLR (Ours)    |    -252.94$\pm$8.53    |    **-173.06**$\pm$7.34   |
> |   RFM-VMLR (Ours)    |    **-260.81**$\pm$6.55    |    -172.53$\pm$3.62    |
>
> Table A: Test NLL (mean $\pm$ standard deviation) computed over 5 runs (lower is better). The number of function evaluations is set to 1000.  The sizes of SPD matrices in BCIC-IV-2a and MAMEM-SSVEP-II are $22 \times 22$ and $8 \times 8$, respectively.
>
> ### **Results**
>
> Table A reports estimates of NLL on the two datasets. Our proposed MLR layers significantly improve RFM on BCIC-IV-2a. We also see that these layers achieve the best performance on both the datasets.
>
> ## **Action Recognition**
> ### **Datasets and Experimental Settings**
> We conduct experiments similar to those in Appendix 1.1.4.
>
> **HDM05 [A5]**
> It has 2337 sequences of 3D skeleton data with $130$ classes. Each frame contains the 3D coordinates of $31$ body joints.  We use the experimental protocol in [A7] in which 2 subjects are used for training and the remaining 3 subjects are used for testing.
>
> **NTU-60 [A6]**
> It has 56880 sequences of 3D skeleton data with $60$ classes. Each frame contains the 3D coordinates of $25$ or $50$ body joints. We use the interaction actions for classification (11 classes) and the cross-view protocol [A6] in which training data come from the camera views 2 and 3, and testing data come from the camera view 1.
>
> We consider a challenging setting in which the input data contain only one of the three coordinates (channels) of human joints. The method in Section 5.1 is used to compute the input data which belong to products of an SPD space and a Siegel space. We focus on comparing our network and SPD neural networks.
>
> ### **Optimization and Hyperparameters**
> We use the same method in Appendix 1.1.2 for optimizing parameters. All networks are trained using cross-entropy loss and Adadelta optimizer for 2000 epochs. The learning rate is set to $1e-2$. We use a batch size of 32 for HDM05, and a batch size of 256 for NTU-60. Experiments are performed using machines with an Intel Core i7-9700 CPU 3.00 GHz.
>
> | Dataset | HDM05 | HDM05 | HDM05 | XView60 | XView60 | XView60 |
> |------------|----------|----------|--------------|-------------|---------------|-------------|
> |              | x-channel | y-channel | z-channel | x-channel | y-channel | z-channel |
> | SPDNet | 38.21$\pm$0.34 | 52.66$\pm$0.36 | 39.25$\pm$0.26 | 64.91$\pm$0.48 | 59.26$\pm$0.40 | 47.36$\pm$0.41 |
> | SPDNetBN | 40.74$\pm$0.30 | 56.82$\pm$0.33 | 42.48$\pm$0.25 | 68.74$\pm$0.46 | 61.97$\pm$0.42 | 50.61$\pm$0.38 |
> | MLR-AI  | 41.32$\pm$0.32 | 59.54$\pm$0.35 | **45.17**$\pm$0.24 | 69.62$\pm$0.50 | 63.87$\pm$0.45 | 50.58$\pm$0.43 |
> | GyroSpd++ | 41.09$\pm$0.37 | 57.32$\pm$0.39 | 43.65$\pm$0.30 | 68.83$\pm$0.51 | 62.12$\pm$0.48 | 49.94$\pm$0.46 |
> | SiegelNet-AFC-ProductQMLR | **42.17**$\pm$0.85 | **66.91**$\pm$0.69 | 44.92$\pm$0.44 | **72.57**$\pm$0.73 | **66.17**$\pm$0.58 | **51.78**$\pm$0.48 |
>
> Table B: Results (mean accuracy $\pm$ standard deviation) computed over 5 runs. XView60 corresponds to the cross-view setting of NTU-60.
>
> ### **Results**
> Results in Table B show that our method surpasses its competitors in most cases. The mean accuracy of  our network using only the y-coordinates (66.91\%) is better than that of SPDNetBN using all the joint coordinates on HDM05 (62.54\% [30]).
>
> **References**
>
> [A1] Ricky T. Q. Chen, Yaron Lipman: Flow Matching on General Geometries. ICLR 2024.
>
> [A2] Clemens Brunner, Robert Leeb, Gernot Müller-Putz, Alois Schlögl, and Gert Pfurtscheller. Bci competition 2008–graz data set a. Institute for Knowledge Discovery (Laboratory of Brain-Computer Interfaces), Graz University of Technology, 16:1–6, 2008.
>
> [A3] Spiros Nikolopoulos. MAMEM EEG SSVEP Dataset II (256 channels, 11 subjects, 5 frequencies presented simultaneously). 2021.
>
> [A4] Mathieu, E. and Nickel, M.. Riemannian continuous normalizing flows. Advances in Neural Information Processing Systems, 33:2503–2515, 2020.
>
> [A5] Müller, M.; Röder, T.; Clausen, M.; Eberhardt, B.; Krüger, B.; and Weber, A. 2007. Documentation Mocap Database HDM05. Technical Report CG-2007-2, Universität Bonn.
>
> [A6] Shahroudy, A.; Liu, J.; Ng, T.-T.; and Wang, G. 2016. NTU RGB+D: A Large Scale Dataset for 3D Human Activity Analysis. In CVPR, 1010–1019.
>
> [A7] Harandi, M.; Salzmann, M.; and Hartley, R. 2018. Dimensionality Reduction on SPD Manifolds: The Emergence of Geometry-Aware Methods. TPAMI, 40: 48–62.

---

> > ### Comment · Reviewer_8RJ3 · 2025-08-04
> > **Response**
> >
> > Thanks for your response. I understand that the rebuttal was limited to 10K characters, so this might have limited the authors from providing a longer answer. Can you reconsider answering my questions/comments? The answers are currently relatively brief and mainly refer to the paper/references, which I understand was due to character limits. Can you explain in detail?
> >
> > Also, thanks for the experiments, can you determine which part of them is new, compared to what you already have in the paper?

---

> ### Author Response · Authors · 2025-08-04
>
> Thank you for your feedback and constructive comments.
>
> And thank for asking again your questions. This gives us an opportunity to provide you and the other reviewers further clarification on our paper.
>
> *Line 72: Can you explain a bit what sort of Riemannian metric induces this distance?*
>
> The distance in line 72 is induced by the following metric [34, 37]:
> \begin{equation}
> ds^2 = 2 \operatorname{Tr}(v^{-1}dxv^{-1}d\bar{x}), \hspace{3mm} x = u + iv \in \mathbb{SH}\_m,
> \end{equation}
> where $\operatorname{Tr}(\cdot)$ is the matrix trace.
>
> Below we summarize some key ideas in [37] that motivate the use of the above Riemannian metric for Siegel spaces.
>
> Recall here that the cross-ratio in Section 2.1 is given by
> \begin{equation*}
> R(x,y) = (x-y)(x-\bar{y})^{-1}(\bar{x}-\bar{y})(\bar{x}-y)^{-1},
> \end{equation*}
> where $x, y \in \mathbb{SH}\_m$. If one considers $R(x,y)$ as a function of $y$, then the second differential of $R(x,y)$ at the point $y = x$ is given by
> \begin{equation*}
> d^2 R(x,y) = \frac{1}{2} dx v^{-1} d\bar{x} v^{-1},
> \end{equation*}
> where $x = u + iv$. Hence
>
> \begin{equation*}
> d^2 \operatorname{Tr}(R(x,y)) = \operatorname{Tr}(d^2 R(x,y)) = \frac{1}{2} \operatorname{Tr}( v^{-1} dx v^{-1} d\bar{x} ),
> \end{equation*}
> where the first equality is due to the community of the differential and trace operators.
>
> It has been shown in [37] that for any $g \in \operatorname{Sp}\_{2m}$, $R(x,y)$ and $R(g[x],g[y])$ have the same eigenvalues. Thus $\operatorname{Tr}(R(x,y)) = \operatorname{Tr}(R(g[x],g[y]))$ and therefore $\operatorname{Tr}( v^{-1} dx v^{-1} d\bar{x} )$ is invariant under the group action of $\operatorname{Sp}\_{2m}$ on $\mathbb{SH}\_m$. In [37], the given differential form $ds^2$ has been proposed as the Riemannian metric (also referred to as Symplectic metric) for Siegel spaces which is a $G$-invariant metric as shown above.
>
> *Line 84: I'm a bit confused; can we obtain any RSS via some G and some K?*
>
> We recall here that in our paper, RSS refer to symmetric spaces of noncompact type or symmetric spaces. One cannot obtain a symmetric space via arbitrary Lie groups $G$ and $K$. However, if $G$ is a connected noncompact semisimple Lie group with finite center, and $K$ is a maximal compact subgroup of $G$, then one can obtain a symmetric space $X$ as:
> \begin{equation*}
> X := G/K := \\{ x = gK | g \in G \\}.
> \end{equation*}
>
> *Definition 3.2: How do you define for elements of the group? Are those coming from matrix representations of a Lie group?
> How do you define log in the same definition?*
>
> Let us focus on the particular case of Siegel spaces. In Definition 3.2, the element $g$ is determined via the map $\phi(\cdot)$ in Section 2.1, that is,
> \begin{equation*}
> g = \phi(x) = \begin{bmatrix} v^{\frac{1}{2}} & uv^{-\frac{1}{2}} \\\ \mathbf{0} & v^{-\frac{1}{2}} \end{bmatrix},
> \end{equation*}
> where $x = u + iv$. The element $h$ is determined via the map $\phi(\cdot)$ in the same way ($h = \phi(y)$).
>
> The function $\log(\cdot)$ computes the logarithm of its input matrix.
>
> In our initial rebuttal, the experiments on Riemannian generative modeling are new, while the experiments on action recognition follow those in Appendix 1.1.4 with a different experimental setting (all the joints and their coordinates are used for computing input covariance matrices, thus leading to much higher-dimensional input matrices) and more challenging datasets.
>
> Please let us know if this answers your questions so that we can provide further clarification.

---

> > ### Comment · Reviewer_8RJ3 · 2025-08-05
> > **Response**
> >
> > Thanks for your clarification and response. I'm happy to increase my score, as I feel the authors provided nice explanations during the rebuttal. But before that, can you briefly list the changes you plan to make to answer my major comment? I mean a specific list of changes (itemized, short sentences). I will update my score right after that.

---

> > > ### Author Response · Authors · 2025-08-05
> > >
> > > Thank you for your feedback and constructive comments.
> > >
> > > To address the major concerns of the reviewer, we plan to make the following changes to the paper:
> > > - Give the expression for the Riemannian (Symplectic) metric in Section 2.1 (in the paragraph "Riemannian Distance", right before the expression for the distance $d\_{\mathbb{SH}}(\cdot,\cdot)$). Also, add some explanations of this metric to Appendix 3.2 which provides more details on concepts related to Siegel spaces.
> > > - Give the explicit expressions for the elements g and h (via the map $\phi(\cdot)$) right before Definition 3.1. This will clarify these elements in both Definitions 3.1 and 3.2.
> > > - Add the new experimental results on action recognition to Appendix 1.1.4 and create Appendix 1.3 for the experimental results on Riemannian generative modeling.
> > >
> > > We hope these changes can address the major concerns of the reviewer. Please let us know if any further changes should be made to improve our paper.
> > >
> > > Thank you again for the discussion and constructive comments.

---

> > > > ### Comment · Reviewer_8RJ3 · 2025-08-07
> > > >
> > > > Thanks, I updated my score from 2 to 4, conditioned on applying the list of promised changes to the paper.

---

> > > > > ### Author Response · Authors · 2025-08-08
> > > > >
> > > > > Thank you for confirming that our answers have addressed your concerns. Thank you again for your thorough review and constructive feedback.

---

### Decision · Program_Chairs · 2025-09-17

**Decision:**

Accept (poster)

**Comment:**

The paper proposes an approach for building discriminative neural networks on Siegel spaces, a family of Riemannian Symmetric Space (RSS).
In particular, the authors show how to build multiclass logistic regression (MLR) and fully-connected (FC) layers for Siegel neural networks, using the quotient structure of those spaces and the notation of vector-valued distance on RSS. Numerical experiments are provided to illustrate the proposed theoretical framework.

The paper has ratings: Borderline Accept, Accept, Borderline Accept, Accept.

Strengths: Reviewers all appreciate the theoretical contributions of the proposed framework.

Weaknesses: Reviewer ahAK would like to see a better motivation via practical applications, i.e. an application that can only be tackled by Siegel Neural Networks. As pointed out by Reviewers ahAK and qfip, the distance in Definition 3.12 is only an upper bound of the actual point-to-hyperplane distance based on vector-valued distance. The authors have said they would clarify this point.

While the paper could certainly be improved, I believe the presented framework would be a novel and interesting contribution to the community. The final recommendation is therefore: Accept

The authors are expected to revise and incorporate the comments made by the reviewers into the final version, as promised in the rebuttal.